# Zinc-Mediated Template Synthesis of Hierarchical Porous N-Doped Carbon Electrocatalysts for Efficient Oxygen Reduction

**DOI:** 10.3390/molecules28114257

**Published:** 2023-05-23

**Authors:** Qianhui Ma, Guifa Long, Xulei Tang, Xiaobao Li, Xianghui Wang, Chenghang You, Wenjun Fan, Qingqing Wang

**Affiliations:** 1Key Laboratory of Water Pollution Treatment and Resource Reuse of Hainan Province, School of Chemistry and Chemical Engineering, Hainan Normal University, Haikou 571158, China; mqh9812@163.com (Q.M.); 18228094599@163.com (X.T.); lixiaobao0797@163.com (X.L.); 2Guangxi Key Laboratory of Chemistry and Engineering of Forest Products, School of Chemistry and Chemical Engineering, Guangxi Minzu University, Nanning 530008, China; gflong@gxmzu.edu.cn; 3Hainan Provincial Key Laboratory of Fine Chemistry, State Key Laboratory of Marine Resource Utilization in South China Sea, School of Chemical Engineering and Technology, Hainan University, Haikou 570228, China; youchenghang@hainanu.edu.cn; 4Collaborative Innovation Center of Chemistry for Energy Materials (iChEM), State Key Laboratory of Catalysis, Dalian National Laboratory for Clean Energy, Dalian Institute of Chemical Physics, Chinese Academy of Sciences, Dalian 116023, China; wjfan@dicp.ac.cn

**Keywords:** oxygen reduction reaction, highly active and stable catalysts, hierarchical porous structures, N-doped carbons, zinc–air battery

## Abstract

The development of highly active and low-cost catalysts for use in oxygen reduction reaction (ORR) is crucial to many advanced and eco-friendly energy techniques. N-doped carbons are promising ORR catalysts. However, their performance is still limited. In this work, a zinc-mediated template synthesis strategy for the development of a highly active ORR catalyst with hierarchical porous structures was presented. The optimal catalyst exhibited high ORR performance in a 0.1 M KOH solution, with a half-wave potential of 0.89 V vs. RHE. Additionally, the catalyst exhibited excellent methanol tolerance and stability. After a 20,000 s continuous operation, no obvious performance decay was observed. When used as the air–electrode catalyst in a zinc–air battery (ZAB), it delivered an outstanding discharging performance, with peak power density and specific capacity as high as 196.3 mW cm^−2^ and 811.5 mAh g_Zn_^−1^, respectively. Its high performance and stability endow it with potential in practical and commercial applications as a highly active ORR catalyst. Additionally, it is believed that the presented strategy can be applied to the rational design and fabrication of highly active and stable ORR catalysts for use in eco-friendly and future-oriented energy techniques.

## 1. Introduction

The depletion of fossil fuels (e.g., oil, coal, and natural gas) and global warming due to greenhouse gas emissions have prompted researchers worldwide to explore clean and sustainable energy sources. Towards this end, many advanced energy technologies have been developed, such as proton exchange membrane fuel cells (PEMFCs), direct methanol fuel cells (DMFCs), metal–air batteries, etc. The oxygen reduction reaction (ORR) is a key progress in these new energy technologies. Owing to its sluggish kinetics procedures, highly active catalysts are required for this reaction. Generally, platinum (Pt)-based catalysts are still the state-of-the-art and most widely used ORR catalysts to now. However, their high cost, limited reserves, and poor stability have hampered the scale-up of their application in practice. Thus, developing highly active and low-cost electrocatalysts to accelerate the ORR is crucial for the scale-up and expansion of the application scope of those advanced and eco-friendly energy techniques mentioned above [1,2,3,4,5,6,7,8,9,10,11,12,13,14,15,16,17,18,19,20].

Carbons, especially N-doped carbons, have received intensive attention and became among the hottest topics in the area of new energy thanks to their advantages such as high ORR performance, low cost, abundant resources, high stability, etc. The past several years have seen great progress in the design of carbon-based ORR catalysts. Despite this, their performances still fail to fulfill the practical requirements for use.

For N-doped carbons, pyridinic and graphitic N species are generally regarded as being ORR-active [21,22,23,24,25]. Thus, appropriately increasing pyridinic and graphitic N content is expected to facilitate the enhancement of their ORR performance by creating more active sites. Unlike graphitic N, which can be only formed during the high-temperature pyrolysis procedures, pyridinic N can stably exist in numerous compounds in the form of a pyridine ring [4,26]. During the pyrolysis procedure, these pyridinic N (or pyridine rings) can be further converted into a graphitic N species [4]; this supplies an opportunity to facially increase the content of these two active N species in carbons. In addition to pyridinic and graphitic N species, Zn-N and Zn-O were recently discovered to be ORR-active as well [27]. Thus, compounds with a high content of pyridinic N coordinated with Zn^2+^ ions can be the precursors to high-performance ORR catalysts.

In addition to creating active sites, access to mass transport and the accessibility of active sites are also important for ORR catalysts. Towards this end, the creation of hierarchically meso-/microporous structures can be helpful as they not only promote mass transport, but also increase the intensity of exposed active sites, as well as their affinity and accommodation (high surface area) [28,29,30,31]. The hard template method is one of the most efficient approaches to produce porous carbons with exceptional pore structures and pore size distributions [32]. As the structural characteristics of porous carbons largely depend on the properties of the used templates, choosing suitable templates is crucial. SBA-15 has been widely used as the template for the fabrication of hierarchically meso-/microporous carbons thanks to their unique properties like parallel open mesochannels with narrow pore-size distributions, thin pores wall of about 3–4 nm, a large pore volume and high surface areas [33].

Thus, in this work, we demonstrate a zinc-mediated template synthesis strategy for a highly active ORR catalyst with hierarchical porous structures by using 4,4′–bipyridine (Bpy) and SBA-15 as the precursor and mesoporous templates, respectively. The optimal catalyst exhibited high ORR performance in a 0.1 M KOH solution, with a half-wave potential of 0.89 V vs. RHE. Additionally, the catalyst exhibited excellent methanol tolerance and stability. After a 20,000 s continuous operation, no obvious performance decay was observed. When used as the air–electrode catalyst in a zinc–air battery (ZAB), it delivered an outstanding discharging performance, with peak power density and specific capacity as high as 196.3 mW cm^−2^ and 811.5 mAh g_Zn_^−1^, respectively. Its high performance and stability are expected to endow it with potential in practical and commercial applications as highly active ORR catalyst. Additionally, it is believed that the presented strategy can be applied for the rational design and fabrication of highly active and stable ORR performance for those eco-friendly and future-oriented energy techniques.

## 2. Results

### 2.1. Physicochemical Characterization

To confirm the coordination of Bpy and Zn^2+^, the precursors were characterized using X-ray photoelectron spectroscopy (XPS). Figure 1a illustrates the high-resolution N1s spectra of Bpy and Bpy-0.5Zn derived from the XPS results. In the N1s spectra of Bpy, only a single peak was observed at approximately 398 eV; this was assigned to the pyridinic N in Bpy. For the blend of Bpy and ZnCl_2_ (Bpy-0.5Zn), the N1s peak was deconvoluted into two peaks at approximately 399 and 398 eV, respectively, which were assigned to the coordinated (N-Zn) and uncoordinated pyridinic N, respectively. The atomic composition of N1s in Bpy-0.5Zn revealed that N-Zn accounted for 60.3 at% (Figure 1b), suggesting that a large portion of pyridinic N species in Bpy was coordinated with Zn^2+^.

In order to study the influence of the addition of Zn^2+^ and SBA-15 on the thermal stability of Bpy, thermogravimetric analysis (TGA) was conducted. Figure 1c shows the TGA results. From the TGA curve of Bpy, it can be seen that a drastic mass loss was observed when the temperature was above 100 °C. As the temperatures further rose, the mass loss became more serious. When the temperature reached 200 °C, little Bpy residue was detected, which should be attributed to the low melting and boiling points of Bpy (109 and 305 °C, respectively). After introducing ZnCl_2_, the thermal stability of the precursor (Bpy-0.5Zn) was drastically enhanced, with approximately 10% of its mass being preserved even after pyrolysis, suggesting Zn^2+^ addition was able to effectively enhance the thermal stability of Bpy. This was attributed to the coordination between Zn^2+^ and Bpy, a fact confirmed by its XPS results (Figure 1a). As for the precursor with SBA-15 (Bpy-SBA), the mass retention rate reached approximately 10% after pyrolysis. This suggested an enhanced thermal stability of Bpy-SBA, which we suggest was attributable to the confinement effect of SBA-15. In the case of Bpy-0.5Zn-SBA, which contains both Zn^2+^ and SBA-15, the mass retention rate ran as high as 40%, suggesting the synergistic effect of Zn^2+^ and SBA-15 in enhancing the thermal stability of precursors.

To analyze the morphologies and microstructures, the obtained catalysts were observed using scanning electron microscopy (SEM) and transmission electron microscopy (TEM).

The SEM image of NC-0.5 (Figure 2a), obtained without any SBA-15 usage, exhibited an irregular bulk-like morphology. In contrast, NC-0-S and NC-0.5-S, as well as other catalysts prepared from the precursors containing SBA-15 (Figure 2b,c and Appendix A), demonstrated bar-like morphologies identical to those of the SBA-15 templates (Figure 2d). Clearly, these bar-like morphologies were inherited from SBA-15. In addition to bar-like morphologies, SBA-15 also provided catalysts with ordered porous structures, as shown in the TEM images in Figure 2e,f. Additionally, such porous structures enabled catalysts to possess smoother mass transfer, better accessibility and more exposed active sites. To clarify the distribution of various elements in NC-0.5-S, scanning TEM (STEM) and corresponding energy-dispersive X-ray spectroscopy (EDS) mapping procedures were conducted (Figure 2g,h). In the EDS mapping profiles (Figure 2h), C, N, and O signals were observed, suggesting that C, N, and O were homogeneously dispersed in NC-0.5-S (Figure 2g,h). However, in the case of Zn, the signal was weak, suggesting that most of the Zn was removed during the preparation. This was also validated by the XPS and inductively coupled plasma-atomic emission spectroscopy (ICP-AES) results (Appendix A and Appendix A), where the Zn content was measured to be only 0.19 wt%.

To further analyzed the porosity of the obtained catalysts, their N_2_ adsorption–desorption isotherms were recorded and the results are shown in Figure 3a. From the isotherms, we observed that all the three isotherms of NC-0.5, NC-0-S, and NC-0.5-S demonstrated hysteresis loops in the medium pressure region, suggesting that there were mesopores in all the catalysts. Comparing NC-0.5 with NC-0-S and NC-0.5-S, we also found that the adsorbed volume of NC-0.5-S in the low-pressure region was much higher. The pore size distribution of the obtained catalysts revealed that the catalysts derived from the precursors that contained both Zn and SBA-15 had much higher pore densities of around 1 nm, which further confirms the synergistic effect of Zn and SBA-15 in improving the porous structures of the catalysts. From the t-plot results, it can be observed that both the micropore and external surface areas first increased and then decreased when the mass ratio between ZnCl_2_ and Bpy increased from 0.4 to 0.6, which can be attributed to the increasing carbon consumption caused by Zn^2+^ [8,34]. As Zn^2+^ was reduced into metallic Zn by carbon, thus, moderate Zn^2+^ addition helped to create more porous structures by consuming the carbon. As a result, the surface area increased. However, when too much Zn^2+^ was introduced, the consumption of carbon substrate also become more serious. In addition, excessive carbon consumption destroy the as-form porous structures as well, especially microporous structures. Thus, excessive Zn^2+^ addition eventually led to a decrease in surface area. Consequently, NC-0.5-S had the highest surface area of 471.1 m^2^ g^−1^ (Figure 3b), as well as the highest external surface and micropore areas, which are believed to supply more effective mass transfer and offer more exposed active sites.

In order study the atomic surface compositions of the various catalysts, XPS measurements were conducted. Among the obtained catalysts, Bpy-0.5Zn exhibited obvious Zn2p peaks at approximately 1020 and 1045 eV (Figure 4a and Appendix A). However, after pyrolysis and hydrofluoric acid leaching, the Zn2p peaks disappeared; this suggested that most of the Zn species were removed during the preparation. The total N content of the various catalysts derived from the XPS results is shown in Figure 4b. As can be seen, NC-0.5 had the lowest N content among the obtained catalysts at 4.82 at%, while the N content of the catalysts prepared from the precursors containing SBA-15 or Zn were much higher than that of NC-0.5. This suggestedthat SBA-15 and Zn helped to reserve more N content during pyrolysis, an occurrence which was also attributed to the confinement effect and the coordination between Bpy and Zn^2+^. Among the catalysts prepared from the precursors with different degrees of Zn addition, NC-0.6-S had the highest N content of 7.95 at%, vs. 6.62 and 7.14 at% for NC-0.4-S and NC-0.5-S (Appendix A). The high-resolution N1s spectra of NC-0.5, NC-0-S, and NC-0.5-S revealed that there were five N species in catalysts derived from the precursors containing Zn (Figure 4c–e and Appendix A), including oxidized, graphitic, pyrrolic, N-Zn and pyridinic N species. Figure 4f shows the atomic compositions of the five N species in various catalysts. After pyrolysis, NC-0.5-S maintained the highest N-Zn content of 26.8 at% among the obtained catalysts, which were recently discovered to be ORR-active [27,35]. Regarding the oxidized N species that are considered inactive for the ORR, the oxidized N contents in NC-0.5 and NC-0.5-S were much lower than that in NC-0-S, suggesting that Zn addition also assisted the removal of O species in the catalysts.

### 2.2. Electrochemical Characterization

In order to evaluate the ORR performances of the obtained catalysts, the cyclic voltammetry (CV) and linear sweep voltammetry (LSV) curves were recorded, as illustrated in Figure 5a,b. Additionally, the half-wave potentials of the obtained catalysts, derived from their LSV curves, are summarized in Figure 5c. It can be observed from the CV curves that NC-0.5 and NC-0-S exhibited poor ORR performance, with peak potential (E_peak_) values of 0.75 and 0.79 V, respectively. The catalysts prepared from the precursors containing ZnCl_2_ and SBA-15 exhibited significantly higher performances, with E_peak_ values of 0.80, 0.86, and 0.82 V for NC-0.4-S, NC-0.5-S, and NC-0.6-S, respectively. This was also confirmed by assessing their half-wave potentials derived from the LSV curves, suggesting that Zn^2+^ and SBA-15 effectively synergistically enhanced catalyst performance. In particular, NC-0.5-S exhibited the highest performance among the prepared catalysts and outperformed the commercial Pt/C catalyst, with a 50 mV more positive half-wave potential (0.89 vs. 0.84 V). To our knowledge, it is also one of the most active ORR catalysts used for ZABs (Appendix A). In order to evaluate the catalytic kinetic process on our catalysts, Tafel analysis was performed. As illustrated in Figure 5d, NC-0.5-S had the lowest Tafel slope of 36.7 mV dec^−1^, validating that it had the lowest overpotential and fastest kinetics during the ORR compared with other catalysts. To further investigate the catalytic process of our catalysts, we conducted rotating ring-disk electrode (RRDE) measurements in an oxygen-saturated 0.1 M KOH solution at a rotating rate of 1600 rpm and calculated the corresponding peroxide yields, as well as the electron transfer numbers. As illustrated in Figure 5e,f, NC-0.5-S had the lowest peroxide yield (<10%) and the highest electron transfer number approaching 4. This was consistent with the values calculated from the Koutecký—Levich (K–L) slope derived from the LSV curves under different rotating rates [36,37,38] (Appendix A), suggesting that the ORR was accelerated almost completely through an ideal four-electron path in NC-0.5-S.

In addition to high ORR performance, methanol tolerance and stability are also important for ORR catalysts, especially in practical applications. Thus, methanol tolerance and stability of NC-0.5-S were also evaluated using chronoamperometry. From Figure 6, it can be observed that NC-0.5-S also exhibited outstanding methanol tolerance and stability. As shown in Figure 6a, the performance of Pt/C decreased sharply with methanol addition; conversely, no significant changes were observed for NC-0.5-S, suggesting its high tolerance upon the methanol addition. After 20,000 s continuous testing, NC-0.5-S maintained 96.1% of its initial performance, whereas Pt/C lost almost one-fifth of its performance under the same test conditions (Figure 6b).

### 2.3. Theoretical Calculation

In order to investigate the influence of the N dopant on the kinetics of the ORR, four typical models with different N distributions and concentrations were selected (Figure 7a) and studied using the density functional theory (DFT) approach (Appendix A). The obtained Gibbs free energy diagrams (Figure 7b,c) show that the intermediate species adsorbed onto the N-bonded C atoms were significantly more stable than those adsorbed onto C atoms that were surrounded by other C atoms. Moreover, the adsorption energies of the intermediates were found to be lowered upon the introduction of an N dopant (Appendix A), rendering the adjacent carbon atoms less dense in terms of charge, which favoured the adsorption of intermediates.

When a C atom was attached simultaneously to two N atoms, the adsorption energy for the intermediate was further reduced (C2 model). This observation was attributed to the lower electron density of the carbon atom, which was induced by the synergistic effect of the two N atoms. In terms of the reaction kinetics, the calculation results suggested that the rate-determining steps on the C1, C2, and C3 sites were the adsorption and activation of oxygen. Thus, the stable adsorption of oxygen facilitated the ORR. Correspondingly, the ORR overpotentials on the C2 and C3 sites dropped to 1.09 and 1.00 V, respectively, in relation to that of the C1 site (i.e., 1.62 V), suggesting a superior ORR performance for the C3 site. In terms of the C4 site, the rate-determining step was the desorption of *OH, however. Although the lowest intermediate adsorption energy was determined for the C4 site, the corresponding ORR overpotential was 3.2 V, which was the highest value among the four sites examined, thereby revealing that the desorption of the intermediate from the C4 site was difficult. These results suggest that increasing the N content and concentrating the N dopants not only increased the number of active sites but also enhanced the intrinsic activity of the nearby C active sites (C3). Among the models studied, the C3 active site, which was surrounded by four N atoms in a square planar structure similar to that of phthalocyanine or porphyrin, exhibited the highest intrinsic ORR activity.

### 2.4. Battery Test

In order to evaluate the potential of our catalyst for use in practical devices, we fabricated a ZAB using NC-0.5-S as the air–electrode catalyst. For comparison, the ZAB was also fabricated using a Pt/C catalyst. As shown in Figure 8a, NC-0.5-S exhibited a much higher discharging performance compared with Pt/C in terms of a much higher discharging current density and a much higher peak power density of 196.3 mW cm^−2^ vs. 145.5 mW cm^−2^ for Pt/C. After complete discharging, the ZAB based on NC-0.5-S exhibited a specific capacity as high as 811.5 mAh g_Zn_^−1^ (Figure 8b). This was much higher than that of the ZAB based on Pt/C (607.9 mW cm^−2^) and approached the theoretical capacity of ZAB (820 mAh g_Zn_^−1^) [39]. So far as we know, it also ranks amongst the highest values recently reported (Appendix A) [40]. These results further confirm the outstanding performance of NC-0.5-S in practical devices, which, we believe, will make it suitable and potential for practical and commercial applications.

## 3. Materials and Methods

### 3.1. Materials

The materials known as 4,4′–bipyridine (Bpy), zinc chloride anhydrous (ZnCl_2_), potassium hydroxide (KOH), and hydrofluoric acid solution were purchased from Shanghai Aladdin Biochemical Technology Co., Ltd. (Shanghai, China). SBA-15 was purchased from XFNANO (Nanjing, China). Nafion 5 wt% (DuPont D520) was purchased from DuPont Co. (Wilmington, DE, USA). Carbon paper (TGP-H-060) was purchased from Toray industries (Tokyo, Japan). All of the chemicals were of an analytical grade. Additionally, all the materials were utilized directly without additional purification.

### 3.2. Preparation of Catalysts

Figure 9 shows the preparation schematics of the catalysts. First, 0.72 g of Bpy and particular amounts of ZnCl_2_ (with the mass ratio of ZnCl_2_ to Bpy of 0, 0.4, 0.5 and 0.6) were completely dissolved in 25 mL of ethanol. The mixture was vigorously stirred for 30 min, and then 0.3 g SBA-15 was added. The mixture was stirred for another 2 h and filtered, followed by rinsing with ethanol and drying in a vacuum at 60 °C. The obtained precursors were named Bpy-mZn-SBA, where “m” refers to the mass ratio of ZnCl_2_ to Bpy. For comparison, the precursors without Zn or SBA-15 were also prepared and named “Bpy-SBA” and “Bpy-0.5Zn”, respectively.

The obtained precursors were first placed in a tube furnace and then heated at 950 °C in an N_2_ atmosphere for 1 h at a heating rate of 2 °C min^−1^. After it had cooleddown naturally in an N_2_ atmosphere, the powder was collected and leached with a hydrofluoric acid solution (40 wt%) for 8 h at room temperature. This was followed by filtration, rinsing with deionized water and alcohol, and drying overnight in a vacuum at 60 °C. The obtained catalysts were denoted as NC-m-S, where “m” refers to the mass ratio between ZnCl_2_ and Bpy in the precursor and “S” refers to the SBA-15 used. To enable a better comparison, catalysts without Zn or SBA-15 were also prepared and named NC-0-S and NC-0.5, respectively.

### 3.3. Preparation of Working Electrodes

For electrochemical measurements, a glassy carbon electrode (GCE, ⌀ 5 mm) was used as the working electrode substrate. Before every measurement, the GCE was cleaned with ethanol in an ultrasonic bath, polished with α-Al_2_O_3_ slurry (50 nm) on a micro cloth, and rinsed with DI water.

For catalyst ink preparation, 5.0 mg of catalyst and a 1 mL Nafion ethanol solution (0.25 wt%) were thoroughly mixed under ultrasound conditions for about 30 min. Additionally, 20 μL slurry was then coated onto the GCE and dried under an infrared lamp. The catalyst loading on the GCE was calculated to be 0.5 mg cm^−2^. For comparison, a commercial Pt/C (20 wt%, Johnson Matthey Corp.) catalyst was also used for ORR (Pt loading: 0.1 mg_Pt_ cm^−2^).

### 3.4. Preparation of Air Electrode for ZAB

For air electrode preparation, Toray carbon paper (TGP-H-060) was used as the substrate. A diffusion layer was formed by painting a suspension of carbon black (XC-72R) and polytetrafluoroethylene (PTFE) onto one side of the carbon paper. During the preparation, the mass ratio between carbon black and PTFE was fixed at 3:2.

For catalyst layer fabrication, a catalyst ink was first prepared through the same procedure described in the working electrode preparation section. The ink obtained was painted onto the other side of the carbon paper and dried under an infrared lamp. The catalyst loading for NC-S900 was calculated to be 1.0 mg cm^−2^. For comparison, Pt/C was also used to fabricate an air electrode through the same procedures (Pt loading: 0.2 mg_Pt_ cm^−2^).

### 3.5. Characterization

The morphologies and nanostructures of the catalysts were observed using scanning electron microscopy (SEM) and transmission electron microscopy (TEM). The SEM images were obtained on a JSM-7100F field emission scanning electron microscope (JEOL, Tokyo, Japan), with an acceleration voltage of 5 kV. Additionally, the TEM was operated on a JEM-2100 transmission electron microscope (JEOL, Tokyo, Japan) at an acceleration voltage of 200 kV. The surface atomic composition of various catalysts and precursors were studied using the X-ray photoelectron spectroscopy (XPS) on an ESCALAB 250 X-ray photoelectron spectrometer (Thermo-VG Scientific, Waltham, MA, USA). During the XPS measurements, the X-ray electron spectra were excited by monochromatized AlKα radiation. The N_2_ adsorption–desorption isotherms were recorded using a Tristar II 3020 automatic surface area and pore analyser (Micromeritics, Atlanta, GA, USA) at 77 K. The thermal stabilities of various precursors were evaluated using thermogravimetric analysis (TGA) on a Q600 thermal gravimetric analyser (TA Instrument, Newcastle, DE, USA). During the test, N_2_ was used as the protection gas. The temperature range used was 25–900 °C, and heating rate is fixed at 10 °C min^−1^. The Zn contents in Bpy-0.5Zn and NC-0.5-S were evaluated using inductively coupled plasma-atomic emission spectrometry (ICP-AES) on an Agilent 720ES inductively coupled plasma atomic emission spectrometer (Agilent, Santa Clara, CA, USA).

### 3.6. Electrochemical Measurements

In order to evaluate catalysts’ electrochemical performance, a three-electrode glass cell was used. Before every measurement, the KOH solution (0.1 M) was saturated with O_2_ (99.999%) for at least 30 min. All the electrochemical measurements were carried out at room temperature on an Interface 1010B electrochemical workstation (Gamry, Warminster, PA, USA), coupled with a rotating ring-disk electrode (RRDE) system (Gamry, Warminster, PA, USA). During the measurements, an Hg/HgO/KOH (1M) (Gaoss Union, Wuhan, China) and a graphite stick were used as the reference and counter electrodes, respectively. All the potentials, initially measured versus Hg/HgO/NaOH (1M) (Gaoss Union, Wuhan, China), were converted into the ones assessed versus reversible hydrogen electrode (RHE) according to E_vs_._RHE_ = E_vs_._Hg/HgO_ + E^θ^_Hg/HgO_ + 0.059 pH. The E^θ^_Hg/HgO_ value was 0.098 V. All the current densities were normalized to the GCE’s geometric area (0.1964 cm^2^).

The linear sweep voltammetry (LSV) measurements were conducted at a scan rate of 5 mV s^−1^ in an O_2_-saturated 0.1 M KOH solution at a rotating rate of 1600 rpm.

The electron transfer number per oxygen molecule involved was first calculated based on the Koutecký–Levich (K–L) equations following [36,37,38]:*J*^−1^ = *J*_L_^−1^ + *J*_K_^−1^ = *B*^−1^*ω*^−1/2^ + *J*_K_^−1^
*B* = 0.62*nFC*_0_*D*_0_^2/3^*ν*^−1/6^
*J*_K_ = *nFκC*_0_

The rotating ring-disk electrode (RRDE) measurements were also conducted at a rotating rate of 1600 rpm in an O_2_ saturated 0.1 M KOH solution using a glassy carbon disk with a Pt ring, which was biased at 1.42 V (vs. RHE). The peroxide yields (*η*) and the electron transfer number (*n*) per oxygen molecule were calculated based on the following equations:*η* = 200*I*_r_(N*I*_d_ + *I*_r_)^−1^
*n* = 4*I*_d_(*I*_d_ + *I*_r_N^−1^)^−1^
where *I*_r_ and *I*_d_ refer to the ring and disk currents, respectively, and N is the collection efficiency, which was confirmed to be 0.36 based on the reduction of K_3_Fe(CN)_6_.

For the Tafel plots, the kinetic current was calculated from the mass-transport correction of the RDE using the following equation [4,5,41]:*I*_k_ =∣*I*_L_*I*(*I*_L_ − *I*)^−1^∣
in which *J* is the measured current density; *J*_K_ and *J*_L_ are the kinetic and diffusion limiting current densities, respectively; *ω* is the angular velocity of the disk (*ω* = 2πN, N denotes the linear rotation rate); *n* is the electron transfer number involved in the reduction of one O_2_ molecule; F is the Faraday constant (F = 96,485 C mol^−1^); *C*_0_ is the bulk concentration of O_2_; *D*_0_ is the diffusion coefficient of O_2_ in the KOH electrolyte; *ν* is the kinetic viscosity of the electrode; *κ* is the electron transfer rate constant; and *n* and *J*_K_ are obtained from the slope and intercept of the K–L plots, respectively. By using the values: *C_0_* = 1.2 × 10^−3^ mol L^−1^, *D_0_* = 1.9 × 10^−5^ cm^2^ s^−1^, and *ν* = 0.01 cm^2^ s^−1^, the electron transfer number (*n*) was calculated.

The methanol tolerance and stability of our catalysts were evaluated by using chronoamperometry measurements. During the measurements, the potential was set to 0.62 V (vs. RHE).

### 3.7. Battery Tests

A home-made ZAB device was designed for the tests. A 6 M KOH solution containing 0.2 M zinc acetate was used as the electrolyte, and a zinc plate was used as the anode. The discharging–charging polarization curves were obtained on an Interface 1010B potentiostat (Gamry, Warminster, PA, USA). The complete discharging experiment was conducted using galvanostatic technology on a BTS-3000 battery testing system (Newware, Shenzhen, China) under a current density of 5 mA cm^−2^. The air electrode and Zn electrode areas exposed to the electrolyte solution were all 1 cm^2^. Additionally, the Zn plate was directly used as the electrode without any pre-treatments. After the electrolyte was added, the ZAB was let stand still for 1 h before it was tested.

### 3.8. Theoretical Calculations

All the calculations were carried out in the Vienna ab initio simulation package (VASP) based on spin-polarized density functional theory (DFT) [42]. The empirical dispersion correction (DFT-D3) method was applied to describe the long-range van der Waals (vdW) interactions in layered materials [43]. The exchange–correlation energy was expressed via a generalized gradient approximation with the Perdew–Burke–Ernzerhof (GGA-PBE) functional, and the projector augmented wave (PAW) pseudopotential was used to represent core electrons effects [44,45]. For all the calculations, the cutoff energy was set to be 500 eV and a Gaussian electron smearing method with σ = 0.05 eV were used. The convergence tolerance for residual force and energy on each atom during structure relaxation was set to 0.05 eV/Å and 10^−5^ eV, respectively. For the model of pure carbon, we used a super cell of lateral size 3 × 3, and the Brillouin zone was sampled with (5 × 5 × 1) Monkhorst–Pack k-points. For the model of N-doped carbon, we used a super cell of lateral size 6 × 6, and the Brillouin zone was sampled with (3 × 3 × 1) Monkhorst–Pack k-points. A vacuum layer of 15 Å along was introduced the z direction to eliminate the spurious interactions between adjacent sheets.

## 4. Conclusions

In summary, a zinc-mediated template synthesis strategy for a highly active ORR catalyst with hierarchical porous structures was presented in this work. The optimal catalyst exhibited high ORR performance in a 0.1 M KOH solution, with a half-wave potential of 0.89 V vs. RHE. Additionally, the catalyst exhibited excellent methanol tolerance and stability as well. After 20,000 s continuous operation, no obvious performance decay was observed. When it was used an air–electrode catalyst in a zinc–air battery (ZAB), it delivered an outstanding discharging performance, with peak power density and specific capacity as high as 196.3 mW cm^−2^ and 811.5 mAh g_Zn_^−1^, respectively. Its high performance and stability are expected to endow it with uses potential in practical and commercial applications as a highly active ORR catalyst. Additionally, it is believed that the presented strategy can be applied for the rational design and fabrication of highly active and stable ORR performance for use in eco-friendly and future-oriented energy techniques.

## Figures and Tables

**Figure 1 molecules-28-04257-f001:**
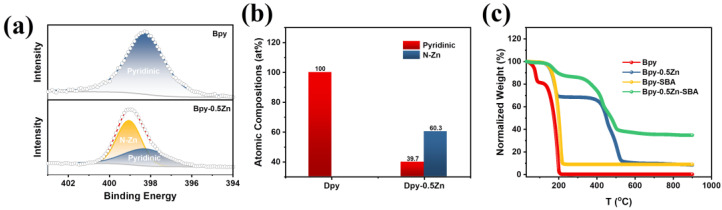
(**a**) N1s spectra of the Bpy and Bpy-0.5Zn: the red line and hollow cycles refer to the raw data and sum of the peaks derived from the deconvolution of the Zn 2p XPS spectra, respectively. (**b**) Atomic compositions of N1s in Bpy and Bpy-0.5Zn. (**c**) TGA results of different precursors.

**Figure 2 molecules-28-04257-f002:**
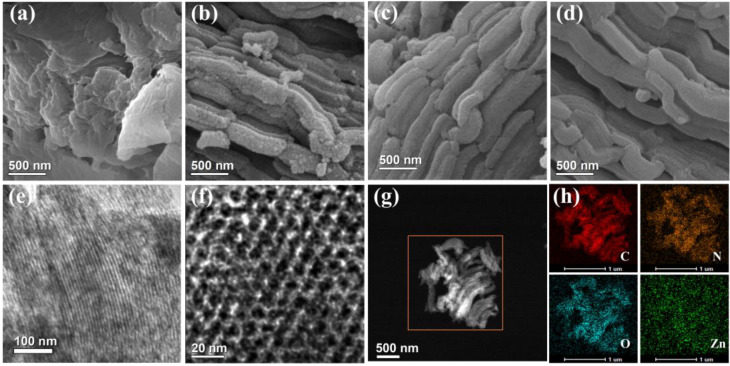
Scanning electron microscopy images: (**a**) NC-0.5; (**b**) NC-0-S; (**c**) NC-0.5-S; (**d**) SBA-15 templates. (**e**) TEM image for NC-0.5-S. (**f**) High-resolution TEM image for NC-0.5-S. (**g**) STEM image of NC-0.5-S. (**h**) EDS mapping of C, N, O, Zn in NC-0.5-S.

**Figure 3 molecules-28-04257-f003:**
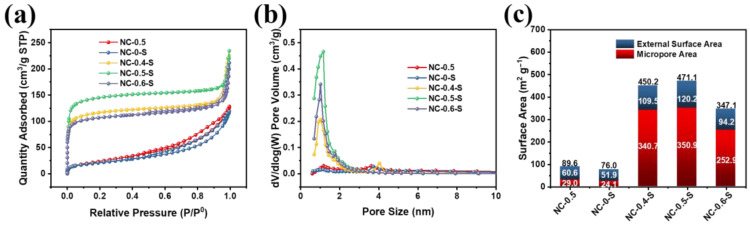
(**a**) N_2_ adsorption–desorption isotherms for NC-0.5, NC-0-S, and NC-0.5-S; (**b**) pore size distributions for obtained catalysts; (**c**) external surface area, micropore area derived from the t-plots, and BET surface areas for various catalysts.

**Figure 4 molecules-28-04257-f004:**
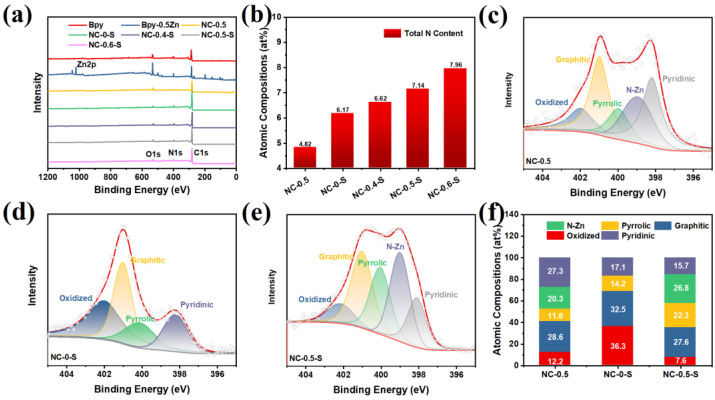
(**a**) XPS spectra for various catalysts. (**b**) Total N content of various catalysts. High-resolution N1s spectra: (**c**) NC-0.5, (**d**) NC-0-S, and (**e**) NC-0.5-S. The red line and hollow cycles refer to the raw data and sum of the peaks derived from the deconvolution of the N1s XPS spectra, respectively. (**f**) Atomic compositions of N1s in NC-0.5, NC-0-S, and NC-0.5-S.

**Figure 5 molecules-28-04257-f005:**
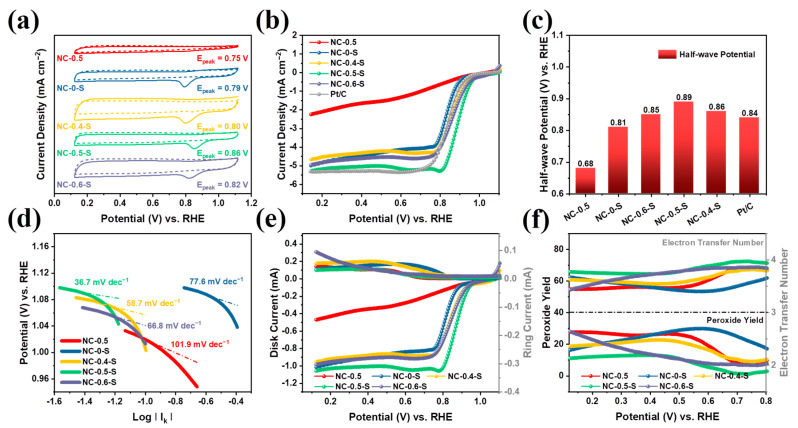
(**a**) CV curves of obtained catalysts in N_2_ or O_2_ saturated 0.1 M KOH solution. (**b**) LSV curves for various catalysts in oxygen saturated 0.1 M KOH solution at a rotating speed of 1600 rpm. (**c**) Half-wave potentials derived from the LSV curves. (**d**) Tafel plots. (**e**) RRDE measurements results. (**f**) Peroxide yields and electron transfer number of obtained catalysts, calculated from the results of RRDE measurements.

**Figure 6 molecules-28-04257-f006:**
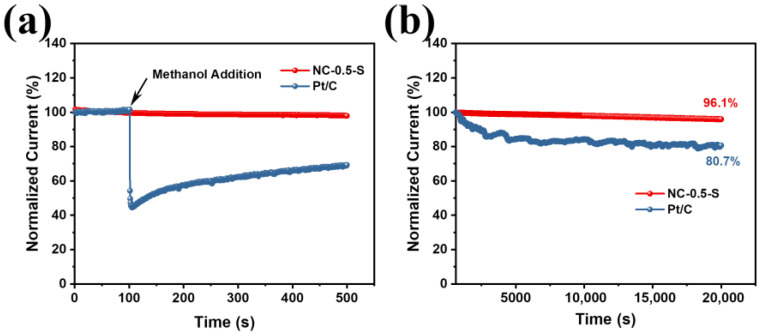
Chronoamperometry measurements results: (**a**) methanol tolerance measurement results, where the arrow indicates the methanol addition; (**b**) continuous i–t curves for NC-0.5-S and Pt/C. During the test, the i–t curves were recorded at 0.62 V (vs. RHE) for both methanol tolerance and stability evaluation.

**Figure 7 molecules-28-04257-f007:**
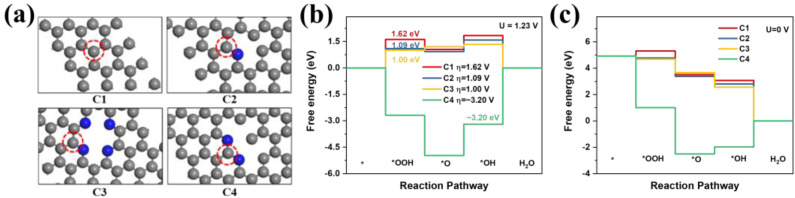
DFT results for the ORR activities of the different carbon sites. (**a**) The various N-doped carbon-type configurations examined in this study (gray spheres = carbon, blue spheres = nitrogen). The carbon atoms denoted by red circles represent the active sites under investigation. (**b**) Free energy diagrams for the ORR pathways on the C1, C2, C3, and C4 sites at U = 1.23 V. (**c**) Free energy diagrams for the ORR pathways on the C1, C2, C3, and C4 sites at U = 0 V.

**Figure 8 molecules-28-04257-f008:**
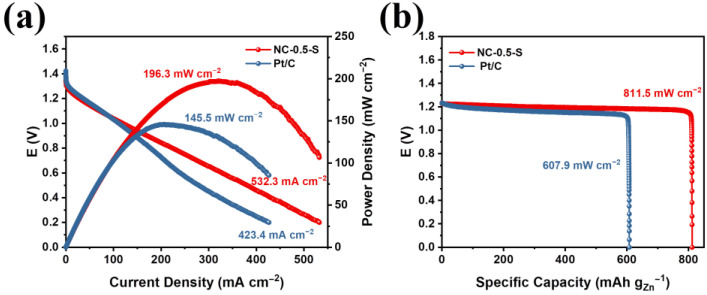
(**a**) Discharging and corresponding power density curves of ZABs using NC-0.5-S and Pt/C as the air–electrode catalysts. (**b**) Complete discharging curves and corresponding specific capacities of ZAB using NC-0.5-S and Pt/C.

**Figure 9 molecules-28-04257-f009:**
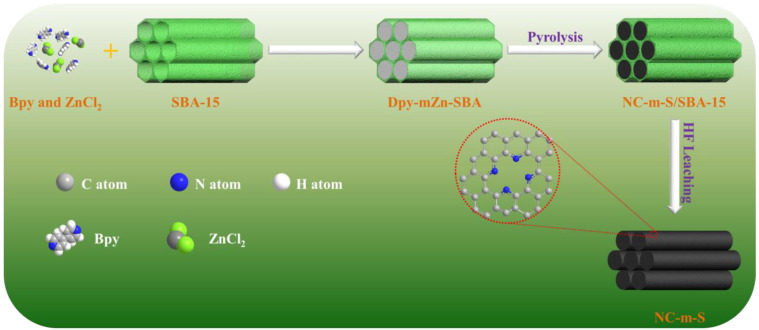
Schematics of the catalyst preparation.

## Data Availability

Not applicable.

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
