# Peer review of "Zinc-Mediated Template Synthesis of Hierarchical Porous N-Doped Carbon Electrocatalysts for Efficient Oxygen Reduction"

_molecules, 2023, doi:10.3390/molecules28114257_

Round 1
Reviewer 1 Report
The authors reported zinc-mediated template synthesis of hierarchical porous N-2 doped carbon electrocatalysts for efficient oxygen reduction. The optical catalyst exhibited high ORR performance in 0.1 M KOH solution, with a half-wave potential of 0.89 V, vs. RHE. This work can be considered for acceptance after the following revisions.
1. The bar in Figure 2a-f should be labeled.
2. The pore size distribution should be presented.
3. To deeply understand the mechanism, theoretical calculations should be performed.
4. The ORR and ZAB performances should be compared with those reported previously in a table.
5. The TEM image of the catalyst after stability test should be given to further characterize the used catalyst in detail.
6. The layout of the figures should be beautified.
7. The recent progress of ORR and ZAB should be discussed in the introduction. Relevant references in this aspect include but not limits to: Molecules 2022, 27(3), 670; Chin. J. Catal., 2023, 46, 48; Molecules 2022, 27(24), 8644; Nano Res., 2023, 16, 2218.
Reviewer 2 Report
Authors present a novel zinc-mediated template synthesis strategy for a highly active ORR catalyst with hierarchical porous structures. Thorough electrochemical and physical characterisation is presented with good primary Zn-air battery performance. This is an original work that should appeal to the readers of the journal Molecules. However, in the current form there are several deficiencies that would need addressing prior to the publication of the paper.
Comments:
1. Page 1, Introduction. the recent paper by Ingawale et al. should be cited and compared what is different in the present investigation: https://doi.org/10.1016/j.cattod.2020.11.016
2. Page 2, line 80. The references for the N1s peak values at 398 eV and also to 399 eV should be provided in the XPS spectra from previous investigations. Also, authors should comment on the proposed presence of Zn-Nx ORR active sites in their catalysts, see https://doi.org/10.1016/j.cattod.2020.11.016
3. Page 3, Figure 1c. Why is there the total weight loss of Bpy sample at 200°C, while the boiling point of Bpy is stated to be by 100°C higher at 305 °C?
4. Supplementary Material, Table S1. What material is Dpy? Should it be Bpy?
5. Page 3, line 122. ICP-AES and XPS data are in conflict for NC-0.5-S as the Zn content was measured to be only 0.017 wt% by ICP-AES, while XPS shows 0.12 at%, which roughly translates to 0.60 wt%. Therefore, the tests should be repeated, work protocols controlled and values corrected.
6. Page 8, Zn-air battery testing. Were the open circuit voltage values recorded, what they were?
7. Page 8, Zn-air battery testing. A recent review paper should be cited (see doi below) and the presented ZAB results should be compared to the other transition metal-based carbon electrocatalysts for Zn-air battery air electrode reported in the literature (see Table 1) in the paper: https://doi.org/10.1016/j.coelec.2023.101229
8. Page 8, line 238. Authors should comment the use of theoretical capacity of ZAB (835 mAh gZn-1) as reviwer knows only the considerably more widespread value of 820 mAh gZn-1.
8. Supplementary Material, Table S2. There is no reference at all to the presence of this table within the manuscript text.
9. Page 9, line 251. Please provide the product number for 0.5wt% Nafion solution as this is very uncommon.
10. Page 9, line 265. How was the pyrolysis temperature of 950°C chosen?
11. Page 9, line 267. What were the acid leaching conditions (e.g. time, temperature, concentration)?
12. Page 10, line 278. Diameter symbol should be ⌀.
13. Page 10, line 282. Why here was different 0.25wt% Nafion solution used?
14. Page 10, line 285. What was the Pt loading on the GC electrode?
15. Page 10, line 285. What was the Pt loading on the ZAB air electrode?
16. Page 11, line 325. What was the value for EθHg/HgO and where it was obtained?
17. Page 11, lines 346-352,
a. What was the air electrode area exposed to the electrolyte solution?
b. What was the Zn electrode area exposed to the electrolyte solution?
c. How was the Zn electrode pre-treated?
The English throughout the paper should be thoroughly revised by a professional. For example, wrong form as optical catalyst is used in the paper instead of optimal catalyst.
Reviewer 3 Report
In this manuscript, Wang and co-authors used the zinc-mediated template to synthesize N-doped carbon catalysts with excellent activity and stability for ORR in alkaline medium. For example, NC-0.5-S shows a high half-wave potential of 0.89 V versus RHE, which is better than those of commercial Pt/C electrocatalysts. Additionally, this catalyst shows high stability and preeminent methanol tolerance, which makes NC-0.5-S a promising catalyst for oxygen electrocatalysis in zinc-air battery applications. Those results could be interesting for the community, and can be accepter after the following issued to be addressed:
Q1:Traditional templates (SBA-15, SiO2, F127, etc.) have been widely applied to introduce mesopores into carbons. In the introduction part did not discuss the importance of the construction of carbon by hard templating processes. It is good to add information about which material is mainly synthesized and what physical properties it has.
Q2:To our knowledge, in the process of synthesizing materials, different carbonization temperatures will inevitably affect the existing form of N in the carbon matrix (SmartMat., 2021, 2, 154-175; Green Energy & Environment, 2022, 7, 1084-1092). Why did the authors choose 950 ℃ as the carbonization temperature? This information was not mentioned.
Q3:Because the signal strength of the XPS data in Figure 4 (c) is insufficient to perform deconvolution, this section is also unscientific.
Q4:As the pore structure of catalysts plays a vital role in the charge-transfer process, it is suggested that the author give detailed parameters of corresponding pore size distributions.
Q5:The author needs to supplement the ORR polarization curves on NC-0.5-S at different rotating speeds from 400 rpm to 2500 rpm, and further analyze the slopes of Koutecky-Levich (K-L). Some typical works may be helpful and certainly need to be cited, such as Acta Phys. -Chim. Sin., 2021, 37, 2009051; CCS Chem., 2022, 4, 1633-1642; eScience, 2022, 2, 227-234.
Q6:The CV results of the synthesized samples in N2- and O2-saturated electrolytes also need to be provided.
Q7: There are still some errors in this manuscript, such as the incomplete ruler in TEM and SEM images on page 4. In addition, the page numbers of references 15 and 18 are incomplete.
Round 2
Reviewer 1 Report
The authors have revised the manuscript and it is recommended to be accepted.
Author Response
Response: Thanks for your comments.
Reviewer 2 Report
Authors have answered to all of the reviewer’s comments and updated the manuscript accordingly. The scientific content is now suitable for publication. However, English language in the manuscript has not been improved at all.
Reviewer found the very same mistake again that he pointed out in the first revision: For example, wrong form as optical catalyst is used in the paper instead of optimal catalyst.
As additional comment here, the optical properties of the catalyst have not been described or studied in the paper in any way. This is just a misspelling of the word optimal and can therefore create confusion for the readers.
Also, authors have introduced new linguistic problems into the manuscript during the revision.
For example, page 2, line 58: In addition to pyridinic and graphitic N species, Zn-N and Zn-O were recently discovered to be ORR as well[27].
Correct version would be: In addition to pyridinic and graphitic N species, Zn-N and Zn-O were recently discovered to be active for ORR as well[27].
These are just a few examples, the language throughout the manuscript would need revision.
There have been no corrections to the English language done. The paper can be accepted after the thorough revision of the English throughout the manuscript.
Author Response
Response: Thanks for your comments. We apologize for the mistakes, and we have corrected them accordingly. Also, we have checked the manuscript and hope it will be satisfied now.